# Designing a Clinical Trial with Olfactory Ensheathing Cell Transplantation-Based Therapy for Spinal Cord Injury: A Position Paper

**DOI:** 10.3390/biomedicines10123153

**Published:** 2022-12-06

**Authors:** Ronak Reshamwala, Mariyam Murtaza, Mo Chen, Megha Shah, Jenny Ekberg, Dinesh Palipana, Marie-Laure Vial, Brent McMonagle, James St John

**Affiliations:** 1Menzies Health Institute Queensland, Griffith University, Southport, QLD 4222, Australia; 2School of Pharmacy and Medical Sciences, Griffith University, Southport, QLD 4222, Australia; 3Clem Jones Centre for Neurobiology and Stem Cell Research, Griffith University, Brisbane, QLD 4111, Australia; 4Griffith Institute for Drug Discovery, Griffith University, Brisbane, QLD 4111, Australia; 5Gold Coast University Hospital, Southport, QLD 4215, Australia

**Keywords:** neurosurgery, olfactory glia, translational health research, regenerative medicine, biomedical engineering

## Abstract

Spinal cord injury (SCI) represents an urgent unmet need for clinical reparative therapy due to its largely irreversible and devastating effects on patients, and the tremendous socioeconomic burden to the community. While different approaches are being explored, therapy to restore the lost function remains unavailable. Olfactory ensheathing cell (OEC) transplantation is a promising approach in terms of feasibility, safety, and limited efficacy; however, high variability in reported clinical outcomes prevent its translation despite several clinical trials. The aims of this position paper are to present an in-depth analysis of previous OEC transplantation-based clinical trials, identify existing challenges and gaps, and finally propose strategies to improve standardization of OEC therapies. We have reviewed the study design and protocols of clinical trials using OEC transplantation for SCI repair to investigate how and why the outcomes show variability. With this knowledge and our experience as a team of biologists and clinicians with active experience in the field of OEC research, we provide recommendations regarding cell source, cell purity and characterisation, transplantation dosage and format, and rehabilitation. Ultimately, this position paper is intended to serve as a roadmap to design an effective clinical trial with OEC transplantation-based therapy for SCI repair.

## 1. Introduction

Spinal cord injury (SCI) is a devastating life-altering condition and there are currently no effective treatments. The loss of sensorimotor and autonomic function that follows an injury has an overwhelming effect on the individual, carers, society, and the healthcare system in general. Apart from paralysis, SCI leads to widespread systemic impact including inflammatory reaction, respiratory issues, cardiovascular complications, compromised immunity and bone densities, muscle wasting and several other complications to the individual’s mental, physical as well as financial health. According to a report by Spinal Cure Australia, there are over 20,800 people living with spinal cord injury in Australia currently, and the lifetime cost of healthcare alone is estimated to be AUD 3.3 billion, with the total lifetime socioeconomic burden being over AUD 75 billion [1]. A conservative estimate suggests that a small significant functional recovery in a fraction of the people living with spinal cord injury can result in savings of AUD 3.5 billion, potentially up to AUD 10 billion [1]. This further highlights the urgent need for a clinically available reparative therapy for SCI, as the current standard clinical practice can only provide damage control and strategies to mitigate complications.

To meet this need, several different approaches are under investigation (Figure 1). Briefly, the core mechanism of action for the approaches used at different stages of injuries include (1) a combination of drugs and anti-inflammatory strategies for damage control and mitigating secondary degeneration in immediate and acute phases—which is currently clinically available [2,3,4]; (2) cell transplantation in the acute, or subacute phase to replace lost glia [5,6]; once the paralysis sets in: (3) robotics to simulate motor repairs [7,8,9]; and (4) biological approaches to restore the natural motor, sensory and autonomic function, which can include cell transplantation, immunomodulation, and growth factor supplementation [10,11]. The biological approaches offer the opportunity to replace the lost neural tissue mass and mitigate the damage by shrinking the defect size. Out of all the stem and non-stem cell types explored for cell transplantation, olfactory ensheathing cells (OECs) stand out as promising candidates for neural repair due to their unique properties [12,13]. The OECs are the primary glial cells of the olfactory nerve, where they play a crucial role in replacing up to 1–3% of olfactory neurons daily and guide them to their intended targets in the olfactory bulb throughout life [14]. OECs from the olfactory mucosal tissue are relatively easy to access via intranasal endoscopy without causing long-term issues, making OECs an excellent candidate for clinical translation, as evidenced by numerous pre-clinical animal trials [15,16,17,18,19,20] as well as several safety/efficacy trials conducted in recent decades [21,22,23,24,25,26,27,28,29,30,31].

Despite this promising evidence, the fact that cell transplantation therapy with OECs (or any other cells) has not yet made it to wider clinical use raises some concerns for caution for future clinical trials. To improve translational outcomes from OEC-based cell therapies, there is a need to identify potential barriers to successful translation and to discuss the preclinical-to-clinical translational approaches with an appreciation of fundamental biological properties of OECs. In this position paper, we discuss the distinct aspects of clinical trial design from donor recruitment, clinical-grade cell production to assessment regimes for us to obtain reliable data from OEC transplantation therapies, and to improve the likelihood of neural repair.

## 2. What to Translate to a Clinical Trial?

Clinical trials are typically translated from pre-clinical animal studies, where there is robust evidence of the safety and efficacy of a treatment modality. In recent years, nearly 20% of the cell transplantation-based clinical trials for SCI were conducted around the use of OECs. OECs have been proven safe for transplantation in animal spinal cord injury models and they have shown varying degrees of success at restoring sensory, motor, and autonomic functions following treatments in rodents [18,19,20,32,33,34,35,36,37], canine [15], and primate models [17]. Based on the evidence from pre-clinical studies, OECs are a promising, low-risk therapeutic candidate for clinical translation with a high chance of success.

### 2.1. What Makes OECs Suitable Candidates for the SCI Repair?

OECs are the supporting cells of the olfactory nerve and are present throughout the course of the nerve from olfactory mucosa to the olfactory bulb. In their natural environment, OECs envelope or ensheathe the olfactory axons and provide them with ongoing support. In this way, they are similar to other glial cells such as Schwann cells and astrocytes. However, unlike the other glial cells, OECs have physiologically developed proficiency in migration and phagocytosis to clean up cellular debris following injuries [38,39], promotion, and augmentation of axonal regrowth by secreting neurotrophic factors [40,41], axonal guidance [42], and modulation of inflammatory profile of their immediate vicinity by expressing different specific cytokine profiles and other molecular markers such as macrophage migration inhibitory factor [43,44]. This can be primarily attributed to their intrinsic environment being a region of high turn-over for axonal damage and repair [5].

Importantly, OECs are shown to retain these direct and indirect mechanisms of inducing and supporting axonal repair when they are transplanted to a neural injury site such as the spinal cord injury [5,10,45]. Additionally, unlike other glia such as Schwann cells, OECs are able to interact with the injury site and the reactive astrocytes that form the scar tissue due to their unique heparin sulphate expression profile [45,46,47]. Thus, the OECs show a natural tendency towards axonal repair. Several types of stem cells are also reported to possess similar properties for nerve repair and secretion of neurotrophic factors [48,49,50]. However, OECs have some distinct advantages over the stem cells for their use in cell-transplantation-based therapy. OECs are differentiated and functionally mature cells, and therefore, they do not need to be differentiated into a functional cell type. In some instances, the OECs have been used as a delivery mechanism for therapeutic transgenes in addition to providing the usual nerve repairs in a spinal cord injury site [51]. Another critical advantage from a safety point of view is that the OECs possess negligible-to-no tumorigenic potential, which is a crucial consideration for a cell transplantation therapy [52]. Being non-stem cells, OECs, especially mucosal OECs, do not face the same logistical, ethical, or moral concerns and controversies for their source as stem cells (except for allografted foetal OECs obtained from abortifacients).

Thus, the natural cellular properties supporting nerve repair, the robust safety profile, and remarkable evidence of efficacy in the animal trials make OECs highly suitable for spinal cord injury repair trials.

### 2.2. What about the Trials so Far?

There have been several clinical trials using OECs in different formats and using different modes of administration. While there is sizeable evidence of safety, the evidence of efficacy lacks consistency, and these trials have not progressed beyond phase IIa. The trials have also identified challenges to a successful and widespread translation of the therapy, with various aspects described below.

Since 2005, there have been around fifty clinical trials attempting cell transplantation-based therapies for SCI across the world. Out of these, eleven trials used OECs as their therapeutic cell population and were focussed on feasibility (pilot), safety (phase I), or evidence of efficacy (phase IIa), and were conducted in many different countries such as Australia, Portugal, India, China, USA, and Poland. However, 1 of the 11 trials was a longer-term follow up of a previous trial (having the same patients) [23], and another trial compared the role of intense rehabilitation in SCI patients who received a cell transplant with those who did not [27], and included patients from a separate trial [28]. Thus, for this review, we have considered the trial design and surgical specifics for only nine trials, although we have included the original individual findings of all eleven trials. Additionally, some case studies involving stand-alone reporting of interesting individual cases related to the use of OECs for spinal cord injury repair were reviewed and are discussed. However, the difference must be noted between the patient-focussed case studies, and the clinical trials where the focus is to answer specific clinical research questions (feasibility, safety, and efficacy) by recruiting patients.

### 2.3. Anatomical Origin of Cells Used in Trials

OECs can be obtained either from the olfactory mucosa via a minimally invasive biopsy from within the nasal cavity, or from the outer layer of the olfactory bulb via an invasive procedure from within the cranial cavity. Some researchers prefer harvesting cells from the olfactory bulb as it can lead to higher purity cultures, but there is a higher risk with the procedure and likely to be permanent damage to the integrity of the olfactory bulb, which may impact on the functioning of the sense of smell. Out of the nine human trials, seven used OECs from the olfactory mucosa, of which three trials used OECs isolated and cultured from mucosal tissues [21,28,29], three used autografts of minced whole olfactory mucosa [22,24,25], and one trial used autologous lamina propria (the part of the mucosa after removing mucosal epithelial lining) for transplantation [31]. The other two trials used OECs obtained from the olfactory bulb [26,30]. Table 1 contains a summary of the details regarding cell origin used in each clinical trial.

Interestingly, a few case studies which were performed outside of a formal clinical trial have also tested OEC transplantation. In a popularly reported case, an autologous olfactory bulb (unilateral) was used as the source of OECs because the patient’s mucosa was rendered inaccessible due to extensive nasal polyps [53]. Another recent case report used foetal olfactory bulbs as the source of OECs [54].

Thus, OECs from both anatomical sources have been tried and tested in clinical settings. Pre-clinical trials have indicated that OECs from both sources have comparable reparative properties [55] despite being two distinctly different sub-populations [56]. The transplantation of cells from both the sources has been shown to be feasible and safe in clinical trials, however, determining the purity of mucosal OECs and their purification have been identified as challenges, thus, raising some serious safety concerns and must be addressed, as discussed later.

### 2.4. Autologous vs. Allogenic Cell Source

Due to the accessibility of OECs from the olfactory nerve, most trials have used autologous transplantation. However, treatments that require early intervention for acute spinal cord injury or when large numbers of cells are required may consider allogeneic donor cells. Two trials used foetal OECs as allografts for treatments, one of which progressed to a phase II study. The use of human foetal OECs does raise numerous ethical issues, and thus, if donor cells are required, the use of adult donor cells would avoid many ethical complications.

In addition to the accessibility of cell source, cell survival and integration after transplantation are important considerations when deciding between these two cell source types [5]. Some pre-clinical trials have indicated that the transient survival of OECs may be sufficient to induce neural repair in SCI, and therefore, immuno-incompatibility may not be critical factor, at least not in the rodent models [16]. However, neuroinflammation and further complications that ensue after a graft rejection, make the autograft a more logical and prudent option.

### 2.5. Cell Purity and Characterisation

OECs can be difficult to purify and cultures of OECs often contain large proportions of other cells, such as fibroblasts. In addition, cultured cells can lose the expression of markers and may potentially alter the function.

Perhaps due to these reasons, the trials using olfactory mucosa (three trials) and olfactory lamina propria autografted the small pieces of intact tissues rather than isolated cells. In this way, the cells were retained within their natural niche, but considerable unwanted cells were also transplanted. However, by using intact tissue, there was no way to characterise the treatment cell population or their purities and no means to define the cell numbers used for treatments in each patient.

Interestingly, most of the remaining trials did not comment on the quality or purity of the transplanted cells in each individual patient—rather, an overview of the cell population was given. For example, the foetal OECs were reported to be ~94% pure in one trial [30]. In another phase I trial, where adult autologous mucosal OECs were used, it was reported that >95% cells expressed the GFAP and S100 markers for OECs and 76–88% cells expressed the marker p75NTR [21]. Conversely, in one trial, the acceptable purity threshold was kept as low as 5% and their treatment cell population was defined as the combination of OECs and the olfactory nerve fibroblasts, which are the most common accompanying cells for OECs of mucosal origin [28]. One trial confirmed their cell populations as OECs by visualising their morphology, but not by immunocytochemistry [29]. Specific details of interest regarding cell purity and characterisation are summarised in Table 1.

Thus, there has been no uniform method or threshold for the acceptability of the transplantation cell population in the clinical trials. This may be an important factor responsible for inconsistent efficacy outcomes. Defining a robust characterisation method and setting a high quality but realistic acceptance threshold is a crucial quality control measure moving forward.

### 2.6. Cell Dosage

The amount of cells transplanted, total treatment volume and the mode of transplantation all have a bearing on the outcomes of the surgical intervention [5]. There is a limit to the amount of the cells and total volume of the treatments that can be transplanted safely at the injury site, however, it is also crucial to maintain the critical mass of the treatment for the therapeutic effect to take place.

As mentioned earlier, cell quantity or treatment volumes were not possible to determine for the four trials that used mucosal pieces or lamina propria grafts. The three trials using autologous OECs prepared in the form of single cell suspensions injected the cells in the cord parenchyma adjacent to the injury in varying doses. The injected cell doses varied widely from 12, 24, and 28 million cells in one trial [21], to 1.8, 1.92, and 21.2 million in another trial (with 30,000–200,000 cells/µL concentration) [28]. In the third trial, a fixed dose of 1 million cells in 2 mL suspension was injected [29]. Two trials using foetal OECs used 500,000 cells in a total volume of 5 µL [26], and a fixed dose of 1 million cells in a total volume of 50 µL [30], respectively. In the case reports mentioning cell suspension injections, 500,000 cells in 48 µL [53] and 1 million cells in 60 µL volume [54] were injected. Thus, the total number and volume of cell suspension injections varied considerably. Surprisingly, the total injection volumes varied from 5 µL to 2 mL in different trials, however, none of the trials reported any links between the treatment volume and final outcomes or adverse events. One complicating factor in linking the dose and outcomes is that cell survival may be affected by the volumes or concentrations that are used.

Understandably, there are considerable differences between the transplantation methods depending on the format in which cells are transplanted. The trials using intact tissue transplanted the treatments directly in the injury site. For this, injury site manipulation, cord untethering, and partial scar debridement were necessary. Conversely, the cell suspension injections were made into the cord parenchyma around the injury site, where scar debridement was optional and not necessary. The injections were generally made as low flow-rate micro-injections at multiple locations. In some cases, all the treatments were carried out identically, where all the patients received 1.1 µL injections in a 3 × 5 grid at 4 different depths [21], or a total volume of 2 mL in 6 injections at the caudal end of the injury site [29]; however, in one trial, patients received 60, 64, and 120 µL by the use of 120, 128, and 212 micro-injections with a fixed 0.5 µL volume with a flow rate of 2 µL/min [28]. The same protocol was used to inject 500,000 cells in a 48 µL volume over 96 total micro-injections in a case report by the same team [53]. The procedural details for these cell transplantations are summarised in Table 2. As mentioned, the injections were made in the relatively healthy and intact cord parenchyma around the injury site where each injection is potentially a separate microinjury. In this manner, both types of approaches are markedly different in their execution and carry their own set of risks and benefits. Even though all the trials concluded that their approaches were safe, a novel approach that combines the benefits of both methods while mitigating the risks may enhance the efficacy of the treatments.

### 2.7. Patient Recruitment

The SCI is a highly variable condition where no two injuries are ever the same, which is why the inclusion of different injury types in the clinical trial may impact on the trial outcomes. Injury level, degree of severity or completeness, and time since injury are some critical factors that may influence the clinical outcomes.

All the clinical trials enrolled patients with at least 6 months after the initial SCI. However, one trial specifically included chronic patients with a neurological profile of the injury that had stabilised for at least 6 months. The included patient ages ranged from 16 to 65 for all trials, with most trials including patients aged 18 or above. Thus, the surgical interventions are attempted in “settled” injury sites in adult patients, where the chances of spontaneous regeneration are null clinically. The acute phase injuries are thus far not preferred for the experimental surgical intervention in the trials which are assessing the safety of the procedure in the first instance.

Most trials included patients with American Spinal Injury Association (ASIA) grade A or B, however, one trial also included ASIA-C patients; one trial used the Frankel scale and only included “complete” injuries on the scale (similar to ASIA-A). The neurological level of the injuries varies widely from cervical to low thoracic levels. Thus, the selection of patients apparently favours complete or more severe injuries in most clinical trials, conceivably to avoid doing any harm in the pilot or phase I trials where the safety of the intervention is still being tested. However, a wider variety of injury types must be considered for the efficacy trials in phase II and further.

### 2.8. Trial Design

Three of the clinical trials were designated “Pilot” studies with 5, 7, and 8 patients enrolled, respectively [22,24,29]. The remaining studies were either designated phase I/IIa trials [23,25], or they started with only phase I and progressed to a phase II study with a 3-year follow-up window [26,31]. Thus, depending on the background pre-clinical work that the trials aim to build on, they may be designed as pilot, phase I, or phase I/IIa trials.

### 2.9. Rehabilitation as an Adjuvant Intervention

Rehabilitation following neurotrauma or other sudden accidental central nervous system events (such as stroke) has been recognised as an important intervention for functional recovery. Rehabilitation is even recognised as a stand-alone treatment modality for spinal cord injury. Recent studies also show that sufficient and sustained rehabilitation may be able to induce limited nerve repairs on its own [27]. This makes rehabilitation a great synergistic adjuvant to any cell transplantation treatment.

Importantly, 5 of the 11 trials, as well as both case reports reviewed here, had a component of rehabilitation associated with the cell transplantation protocol, while 1 of the trials assessed neurorehabilitation as the primary intervention [27]. Notably, three of the trials included a pre-operative component of intense rehabilitation ranging from 3 months [28] and 6 months [30] to 8 months (35 weeks) [25]. These trials also had the post-operative intense rehabilitation of 24 months [25] and 21 months [28]; however, information was not specified in the third trial protocol. Another trial included only the post-operative component of rehabilitation lasting from 2.5 to 4.5 months [27], whereas the remaining trial instructed the participants to perform home-based rehabilitation [31]. The dose of the rehabilitation also varied considerably, ranging from an average of 8.5 h a week [27], to nearly 20 h a week (4–5 h per day, 3–5 days a week) [28], with the highest amount reported between 25 and 39 h a week [25].

It is important to note that the trial by Lima et al. included the highest weekly hours and longest total duration of rehabilitation, however, no patient dropouts were mentioned [25]. Conversely, the trial by Chen et al. had a mandatory 6 months of pre-operative rehabilitation included but only 28 of the original 64 participants were able to complete the intensive rehabilitation program; however, only 7 of these 28 patients were able to receive the transplants due to limited resources [30]. Thus, neurorehabilitation, akin to the several other specific parts of the protocol, showed marked differences across different trials. Most commonly, the availability of resources, such as funding and patient adherence, were the determining factors in this regard.

### 2.10. Outcome Measures

Most of the studies focussing on the safety profile of the intervention primarily defined their outcome measure as patient safety, which is monitored by recording any adverse events (AE) or severe adverse events (SAE) as well as any regression of the neurological function/worsening of existing symptoms. The patients were followed up for at least 1-year post-intervention in most trials, with only one exception, where the follow-up period was 6 months. The maximum follow-up period was 48 months in one trial and a few more trials had up to 3-year follow ups.

Almost all the trials reported no SAEs and minimal AEs. However, one trial reported a patient with reduced ASIA sensory grade with pain and tingling over 18 months of follow up, and another trial reported a case of meningitis, cerebrospinal fluid leak, and new occurrence of visceral pain over the follow up of 48 months.

The efficacy of the interventions was assessed using several different clinical assessment tools across different clinical trials as the secondary outcome measures, such as ASIA grading, electrophysiological assays such as electromyography, somatosensory-evoked potentials (SSEP), motor-evoked potentials (MEP), autonomic function such as bowel and bladder control, imaging changes as seen on MRI and diffusion tensor imaging (DTI), monitoring of neuropathic pain, functional independence measure (FIM), activities of daily living rating (ADL), walking index for spinal cord injury (WISCI), modified Ashworth’s scale for spasticity (MAS), and international association of neurorestoratology—spinal cord injury functional rating scale (IANR–SCIFRS). Thus, there are numerous assessments that can be carried out to determine functional and psychosocial outcomes.

### 2.11. Safety Concerns

In addition to the AEs and SAEs reported in the clinical trial outcomes (see Table 1), there are a few further safety concerns. Outside of the reviewed clinical trials and case reports here, there have been additional case reports where improper cell harvesting techniques and/or surgical protocols have resulted in disastrous outcomes for the patients several years after the intervention.

The earliest reported case was that of a young female patient presenting with a cystic mass at the transplantation site 8 years after the intervention [57]. More recently, another case of a 38-year-old male was also reported with a similar presentation, 12 years after the intervention [58]. Both these patients presented with severe back pain and needed a subsequent surgery to remove the mass. Surprisingly, the masses were found to have mucus-producing cells from the respiratory mucosa. Both these unfortunate incidents occurred from the transplantation of mucosal pieces that demonstrate that transplanting unquantified, uncharacterised, and uncultured tissue grafts is not a desirable transplantation approach. This further highlights the importance of having a thorough understanding of the relevant cell types and their physiological properties as well as having robust isolation, purification, and characterisation protocols in place for the transplantation cell population.

### 2.12. Conclusions of the Past Clinical Trials

All the studies unequivocally concluded that the therapy was safe and feasible, and most studies found that the therapy was effective with varying levels of functional recoveries recorded, as summarised in Table 1. However, none of the therapies have progressed beyond this level of clinical testing, which may be due to a combination of limited resources and variable outcomes in the clinical trials. There are numerous potential factors contributing to variable outcomes, which if addressed, could improve the efficacy of the treatment. For example, the past trials transplanted cells either as single cell suspension via injections, or as small pieces of tissue where the quality and quantity of the cells cannot be controlled or confirmed, likely leading to a high variation in outcomes. Trials opting to inject cell suspensions did not address the scar tissue, whereas the trials transplanting mucosal pieces performed varying degrees of partial scar removal. The scar removal was shown to be safe, however, much debate still persists regarding the wisdom of scar removal. One important aspect of the past trials was the use of fixed volume/dose treatments for different injuries. Different injuries with different volumes, shapes, and sizes are likely to respond differently, and thus, the cell dose needs to be tailored to suit the injury.

## 3. How to Design a Clinical Trial?

A clinical trial must be strategically planned around the primary aims of the trial, specifically as to avoid any confounding factors from the skewing trial outcomes. It is also critical to incorporate measures to address and overcome the limitations and issues identified during past clinical trials in order to successfully progress further if the trial’s aims are met sufficiently.

In our proposed approach, we have identified the following strategies to address and overcome these challenges, thereby strengthening the approach for OEC transplantation to repair SCI.

### 3.1. Olfactory Mucosa as a Source of the Cells

It is clear from the clinical trials that the preferred source of autologous OECs is the olfactory mucosa from the nasal cavity. In the only instance where autologous cells were obtained from the olfactory bulb, it was carried out because the mucosa was rendered inaccessible [53]. The OECs of mucosal origin are easily accessible with minimally invasive means using local anaesthesia, or with general anaesthesia if the surgeon prefers to gain deeper access to the nasal cavity. As the nasal mucosa is the only tissue that is involved, there is little risk to the patient during the harvest procedure. From the point of view of the integrity of the sense of smell, harvesting a small region of the nasal mucosa does not affect the ability to smell. In contrast, harvesting OECs from the olfactory bulb requires invasive access into the cranial cavity and is likely to result in considerable damage to the nerve fibre layer of the olfactory bulb, with likely ongoing perturbation to the functional capacity of the sense of smell. Importantly, the mucosal OECs are known to express a unique combination of proteins that are developmentally relevant for nerve repair and are not expressed by the OECs from the bulb [59,60].

### 3.2. Comprehensive Assessment of Cell Purity and Function

To improve the clinical outcomes from OEC-based cell transplantation therapies, there is a need to standardize the cell isolation, expansion, and good manufacturing practice (GMP) of cell production aspects. There is no consensus on the methods for isolating OECs from the olfactory tissues or markers to identify OECs and the other cells obtained in the primary culture [59,61]. The cells obtained from the olfactory mucosa for clinical purposes should be tested at various points during the GMP production process and should be reported for every patient. At a minimum, the starting material, the intermediate expansion product, and the final cellular product must be analysed. Cell assessments should involve an analysis of cell size, shape, morphology, growth characteristics, and the evaluation of cell surface markers. The cells should be analysed for the cell surface marker expression of a combination of positive and negative markers by quantitative immunofluorescence staining or flow cytometry. These assessments will help determine the quantity and purity of the cells, as well as identify subpopulations of cells. For the clinical trial, the research team must define the specified ranges of cell surface marker expression and viability, and this pre-determined acceptance criteria must be used prior to the release of OEC cells for clinical transplantation.

In the previous clinical trials conducted for OECs, the biological activity of the transplanted cells has not been shown prior to cell transplantation. However, the measured biological activity must be related to the intended biological effect as there is the potential for the culturing conditions to alter cell function. Therefore, there is a need for assays to pre-determine the biological effect of OECs as a potential predictor of clinical outcomes and to control the quality of the cell therapy product. However, this can prove to be challenging—it can be difficult to produce large quantities of cells from the olfactory biopsy and diverting cells for functional testing may adversely impact the dosage available for clinical use. The development of assays must, therefore, keep these limitations in mind and use small numbers of cells. Further, the assays should be simple and cost-effective for widespread adoption in clinical practice.

### 3.3. A 3D Construct Is Warranted

Cell integration and survival is one the major factors affecting the outcomes of any cell transplantation-based therapy including OECs. Several factors such as inflammation and hostile milieu of the SCI site adversely affect the ability of OECs to survive and integrate with the injury site. Additionally, the cells have historically been transplanted in a suspension form via an injection in the healthy cord parenchyma around the injury site. These needle tracks essentially inflict further injury to an already injured spinal cord, and there is a limited volume that can be applied to the spinal cord.

To overcome this, some trials in the past have opted to transplant pieces of olfactory mucosa or lamina propria directly in the SCI site, without manipulating the surrounding healthy cord tissue. This provides the transplanted cells their own native tissue scaffolding, and thus, improves their chances of survival and integration; however, this approach has its own significant draw backs. The lamina propria has several other cell types present and the OECs can neither be purified nor quantified prior to transplantation. Thus, unknown quantities and quality of the transplanted cells are transplanted which can be a significant confounding factor leading to widely varying outcomes. The approach also has some associated risks as it involves non-purified tissue autograft. The most significant risk is the possibility of accidentally transplanting tissues collected from sub-optimal sites which may contain respiratory mucosa or cells other than OECs, thus, potentially leading to disastrous outcomes [58].

An alternative cell preparation approach is to create three-dimensional constructs in which cells are embedded within supporting structures or gels [62,63], thus, enabling them to be deposited into the injury site.

### 3.4. Cell Dose and Treatment Volume

As mentioned before, there is a huge disparity amongst the trial with regard to the cell dose and treatment volumes. Assuming the highest reported numbers to be the maximum safe treatment thresholds, the highest used treatment volume was 2 mL and in a separate trial, the highest used cell dose was ~21 million cells. Thus, the maximum dosage that can be performed safely via injections is twenty million cells in 2 mL volume. However, using the alternative 3D preparation approach presents another critical edge in this regard, as the volume/amount of cells can be tailored to suit the size of the cavity, with surgeons able to customise the dosage to fill the cavity with the 3D cell preparation.

### 3.5. Surgical Approach for Transplantation

It has been well established that the scar at the site of SCI is unique in its cellular and molecular make up. The scar also plays a vital role in the protection and stabilisation of the injury site up to a certain point; however, after the injury is stabilised, the scar tissue acts as a physical barrier to any potential repairs. The molecules such as chondroitin sulphate proteoglycans (CSPG) also emit inhibitory signals for the reparative processes.

Thus, strategic manipulation of the scar tissue is crucial for a successful transplantation intervention. Leaving the scar tissue untouched can impede the cellular repairs and axonal growth through the injury zone; however, the removal or over-dissection of the scar can further destabilise the injury site and trigger adverse neuroinflammatory signals. One of the advantages of using OECs is that they can interact and permeate through the dense glial scar of the SCI [64,65], and therefore, a complete or partial removal of scar may not be necessary. We propose that a tactical minimal debridement of the scar tissue aimed at merely mobilising the adhesions and securing an approach to deposit the 3D preparation of cells within the defect would be best suited.

### 3.6. Patient Recruitment Should Be Carried out Based on the Clinical Trial Phase

#### 3.6.1. Complete or Incomplete Injuries

Patient safety is a priority. For this reason, chronic injuries in which the neurological function has stabilised are the safer option for cell transplantation clinical trials at this point. In addition, as most phase I and IIa clinical trials have done in the past, patients with a more complete injury profile should be selected as there is a reduced risk of causing further neurological harm due to the intervention. Similarly, the lower neurological level injuries are best suited to test for the safety, however, a more lenient range can be set for including patients with the different neurological levels of injury depending on access to the patient pool and patient enrolments. It is worth considering that the patients with incomplete injuries have a larger portion of cord tissue spared with less extensive scarring, and therefore, have a higher chance of benefitting from the therapy. Once the use of OEC transplantation has been shown to be safe, patients with less complete injuries could be considered for potential treatment.

#### 3.6.2. Time since Injury

Another aspect for the SCI profile is the time since injury. While ‘the sooner the better’ stands true for regaining neurological function after SCI, there are several factors to be considered in this regard. Neuroinflammation is a complex process which significantly affects the survival and integration of transplanted cells. For some patients, there may be a spontaneous regain of function following an injury, which an early intervention can impede. Depending on the extent of the initial injury, the injury might be deemed too unstable for an invasive experimental intervention such as cell transplantation. Considering all such factors, patients with injuries at least 12 months old should be included for the safety trial. The injury site is fairly stable at this point, and the probability of spontaneous repairs becomes considerably reduced after the first few months of injury. Thus, there is less chance of harm by intervention at this stage. The additional advantage of including chronic injuries is that the enrolment process can be expedited since there is no need for prospective enrolment, which can be time consuming and limited. Once the safety of the cell transplantation treatment is confirmed, acute/subacute injuries can be included in future trials.

### 3.7. Adaptive Trial Design May Be Beneficial

With the clinical translation of such critical therapies, the clinical benefits and urgency must be balanced against patient safety and pragmatism. Thus, we propose that new clinical trials should primarily be aimed at assessing the safety and feasibility despite the abundant evidence supporting the safety of OECs. Although this approach uses the same cells, new purification and identification techniques and functional assessments, as well as the ability to prepare cells in a 3D format, will warrant another phase I trial. However, efficacy can be pursued as a secondary objective and a phase I/IIa trial can be designed accordingly. In the likely event of a successful phase I trial outcome, an adaptive trial design can be employed for the subsequent studies to expedite the further translation. For example, with an adaptive design, the cell dosage and timing can be tested and revised as feedback from the outcomes of the first patients is obtained. In this way, the optimal findings can promptly be re-incorporated in the trial.

### 3.8. Outcome Measures Must Be Selected Strategically

The primary outcome measure must remain to be patient safety as monitored by the adverse events and severe adverse events, as well as the worsening of any existing symptoms. However, for the secondary outcome measures, a wide variety of clinical assessment tools can be used as reported by the past clinical trials. Nevertheless, it is critically important that none of the therapeutic impacts go undetected. Therefore, the clinical assessment tools must be employed strategically to guarantee that the assessments can detect any change in the patients’ condition after intervention, while ensuring that the outcomes are not over-interpreted. This is also necessary for a cost–benefit evaluation. For example, AIS grades, the most commonly used clinical tool, may be too crude to pick up small-scale improvements that can be detected by the spinal cord independence measure (SCIM), minimal clinically important difference (MCID), or a reduction in neurological level of injury (NLI) [11]. A clinically small significant improvement is defined as a four-point improvement in SCIM, which may or may not be reflected in AIS grading at all; however, such a change may drastically improve the quality of life for the patient. Additionally, a conservative economical estimate by the Australian government suggests that a therapy that can consistently yield such a small significant improvement may result in savings of over AUD 3.5 billion. On the other hand, including too many assessment tools can be counterproductive by increasing the costs of the trial and reducing patient adherence or causing distress to the patients.

### 3.9. Rehabilitation Is Crucial for Functional Recovery

The reviewed clinical trials and case reports all definitively conclude that rehabilitation is crucial for functional recovery following the transplantation of OECs. There is some debate regarding if the rehabilitation alone is sufficient for any significant recovery, and that combining the cell transplantation with rehabilitation makes it difficult to link any potential recovery to cell transplantation. However, the aim of this position paper is to derive clinically relevant information for developing therapeutic approaches, and as such, the recommendation is clearly in favour of combining cell transplantation with rehabilitation. The overarching aim here must be to develop an approach that helps clinical patients regain their lost functions, and intense rehabilitation combined with the transplantation of OECs offers the best chance to achieve that.

Ideally, the amount of rehabilitation should be kept uniform (albeit, not necessarily identical) for all participants, where they would spend similar hours on distinct aspects of training such as posture, balance, pre-gait, and gait trainings as well sensory training in each session. The specifics of the sessions, however, such as the weights and intensities of each different training session, should be customised from patient to patient to maintain feasibility to continue over a long time, while avoiding stagnation in their recovery. Having a personalised approach to rehabilitation is critical as each patient’s journey will be different. It is, therefore, advisable to have a minimum target for the amount of rehabilitation, but allow for some flexibility for participants in the trial to adjust their rehabilitation regimen to suit their individual needs and capacities.

Obtaining sufficient funding for rehabilitation has been a major limiting factor in past trials. Several distinct aspects of the prolonged rehabilitation such as accessibility, availability of the rehabilitation at multiple geographic locations, and a balance between site-based as well as home-based rehabilitation programs would rely on access to sufficient funding. Considering the importance of rehabilitation in complementing the cell transplantation, it is critical that the trial is funded sufficiently to enable participants to complete the rehabilitation program.

### 3.10. Prehabilitation Is Indicated

Several of the past trials included a pre-operative neuro-rehabilitation component. Some trials explained the rationale of this intervention, as it is important to see if there was any opportunity for spontaneous recovery. Additionally, pre-operative rehabilitation (or prehabilitation) is likely to prime the participants for post-operative rehabilitation—which is of key importance—prepare them with what they can expect after the treatment surgery, and overall enhance the patient adherence, thus, improving the likelihood of success overall. Therefore, prehabilitation with the same intense rehabilitation regime is indicated.

## 4. Conclusions

Olfactory ensheathing cell transplantation offers a promising therapy for repairing spinal cord injury. Clinical trials of OEC transplantation have shown that it is safe and feasible, however, past clinical trials also highlight challenges that must be overcome to complete a successful and widespread clinical translation of the therapy. Future clinical trials should be designed to incorporate a range of aspects that will increase the likelihood of success. The cells sourced from olfactory mucosa represent the safest clinical approach, however, complete and robust cell characterisation and quality control is necessary to ensure that the appropriate treatment cell population is transplanted. Importantly, transplanting cells in a three-dimensional format is the most suitable way to overcome the adversities of surgically transplanting cells into the spinal cord. While patient recruitment will likely involve people living with chronic injuries, planning for treating acute injuries can be incorporated into an adaptive trial design which can also test changes in dose. Finally, combining cell transplantation with neurorehabilitation provides the best chance of functional recovery for the trial participants.

## Figures and Tables

**Figure 1 biomedicines-10-03153-f001:**
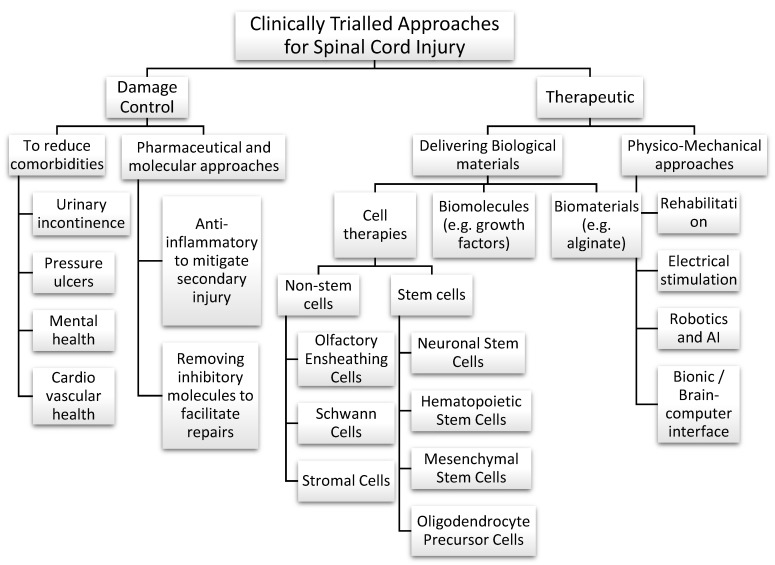
An overview of approaches under exploration for spinal cord injury treatment.

**Table 1 biomedicines-10-03153-t001:** A summary of details of interest regarding clinical trials design and protocols.

Author, Year	Transplantation Format	Cell Characterisation	Purity	Adverse/Severe Adverse Events	Safety Established	Efficacy
Féron et al., 2005 [21]	Isolated mucosal OECs (autograft)	GFAP + S100; p75NTR	>95%; 76–88%	None	Yes	Not assessed
Lima et al., 2006 [22]	Olfactory mucosa pieces (autograft)	N/A	N/A	None	Yes	Modest improvement in ASIA scores
Mackay-Sim et al., 2005 [23]	Isolated mucosal OECs (autograft)	N/A	N/A	Meningitis, CSF leakage, IBS, new visceral pain	Yes	No significant functional improvements
Chhabra et al., 2009 [24]	Olfactory mucosa pieces (autograft)	N/A	N/A	1 syrinx; 1- more sensory loss recovering gradually	Yes	No significant improvements, but statistically significant improvements in SCIM, BDI and ISCIS
Lima et al., 2010 [25]	Olfactory mucosa pieces (autograft)	N/A	N/A	None	Yes	Possible with post-operative rehabilitation
Wu et al., 2012 [26]	Foetal OB OECs (allograft)	GFAP and S100 Immuno-staining		One patient had reduced ASIA sensory with pain and tingling, transient pain resolving with analgesics. No SAEs	Yes	Moderate sensory and spasticity improvements, minimal locomotor improvements
Larson et al., 2013 [27]	Olfactory mucosa pieces (autograft)	N/A	N/A	N/A	Yes	Motor recovery was observed, no sensory improvement. Recovery was not significantly greater compared to the control group
Tabakow et al., 2013 [28]	Isolated mucosal OECs (autograft)	S100, p75NTR	>5%	Some immediate adverse events over post-operative phase, resolving within 3–4 days. No AE or SAE over 1-year follow up.	Yes	2 of the 3 patients improved ASIA scores, third patient had some neurological recovery without ASIA score improvement
Rao et al., 2013 [29]	Isolated mucosal OECs (autograft)	morphology	Not reported	No SAEs	Yes	3/8 patients had substantial sensorimotor recovery; 2/8 had bladder function restored
Chen et al., 2014 [30]	Foetal OB OECs (allograft)	p75NTR, S100 for OECs; S100 for SCs	94%	One patient had fever, No SAEs	Yes	4/5 treated patients showed significant electrophysiological improvements, 5/5 showed some functional improvement
Wang et al. [31]	Olfactory lamina propria pieces (autograft)	N/A	N/A	No SAEs	Yes	Limited functional recovery; 2/8 patients had ASIA score improvement

ASIA = American Spinal Injury Association, BDI = Beck Depression Inventory, ISCIS = International Spinal Cord Injury Scale, SCIM = Spinal Cord Independence Measure, OB = Olfactory bulb, OECs = Olfactory ensheathing cells, AE = Adverse event, SAE = Serious adverse event. N/A = Not applicable.

**Table 2 biomedicines-10-03153-t002:** Procedural detail summary for cell transplantation in the clinical trials using cell suspension injections.

Author, Year	Cells Transplanted	Cell Concentration	Injection Volume	Flow Rate	Number of Total Injections
Féron et al., 2005 [21]	12 million, 24 million, and 28 million, respectively, injected in 3 patients	not mentioned	1.1 µL/injection, ~132 µL total	not mentioned	4 depths in a 3 × 5 grid, both rostrally and caudally = 120 injections
Wu et al., 2012 [26]	500,000 cells	100,000 cells/µL	5 µL	not mentioned	2 injections, 1 each rostrally and caudally from the injury
Tabakow et al., 2013 [28]	1.8 million, 1.9 million, and 21 million cells, respectively, in 3 patients	30,000–200,000 cells/µL	60, 64, and 106 µL, respectively	2 µL/min	120, 128 and 210 injections, respectively, 0.5 µL per injection
Rao et al., 2013 [29]	1 million cells	50,000 cells/µL	2 mL	not mentioned	6 injections total
Chen et al., 2014 [30]	1 million cells	20,000 cells/µL	50 µL	not mentioned	2 injections, 1 each rostrally and caudally from the injury

## Data Availability

Not applicable.

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
