# Peer review of "Designing a Clinical Trial with Olfactory Ensheathing Cell Transplantation-Based Therapy for Spinal Cord Injury: A Position Paper"

_biomedicines, 2022, doi:10.3390/biomedicines10123153_

Round 1
Reviewer 1 Report
This is a very well written paper aimed to serve as a roadmap to design a new clinical trial with olfactory ensheathing cells transplantation-based therapy for spinal cord injury. The olfactory ensheathing cells were among the first, and still the most promising cells to consider for spinal cord injury therapy.
The authors analyze shortcomings of previous studies, and single out all clinically relavant points to improve study design. If any cell transplantation therapy proves to be efficient in improving outcome of spinal cord injury, it is probably the olfactory ensheathing cells. Thus, this position paper is in a way a mission statement from the credible group of authors, both clinicians and scientists, and as such deserves to be published.
Author Response
We thank the reviewer for the comments and note that there are no changes have been suggested.
Reviewer 2 Report
This is an interesting review article discussing olfactory ensheathing cell transplantation-based therapy for spinal cord injury. It is a good writing with concise presentation. The review is also important to other scientists for further developing cell-based therapy for spinal cord injury.
Minor comments:
1. More latest references are needed to be cited in the manuscript.
Author Response
Reviewer comment: More latest references are needed to be cited in the manuscript.
We have added some more recent references to the manuscript:
- Page 2, lines 51-53: 3 references added – 1 reference each after the highlighted text (numbered 4, 6, 9)
- Page 2, line 83: 4 references added (numbered 34, 35, 36, 37)
- Page 3, line 106: 1 reference added (numbered 47)
- Page 12, line 397: 1 reference added (numbered 61)
We have searched the literature again for any more recent clinical trials for inclusion in this manuscript and have found no new publications relevant to this manuscript.
Please note that the “track changes” function does not work with adding references in MS Word, therefore, we have highlighted short parts of the text where the references are added.